# Ribosome Pausing Negatively Regulates Protein Translation in Maize Seedlings during Dark-to-Light Transitions

**DOI:** 10.3390/ijms25147985

**Published:** 2024-07-22

**Authors:** Mingming Hou, Wei Fan, Deyi Zhong, Xing Dai, Quan Wang, Wanfei Liu, Shengben Li

**Affiliations:** 1Shenzhen Branch, Guangdong Laboratory of Lingnan Modern Agriculture, Genome Analysis Laboratory of the Ministry of Agriculture and Rural Affairs, Agricultural Genomics Institute at Shenzhen, Chinese Academy of Agricultural Sciences, Shenzhen 518120, China; houmingming_mbb@126.com (M.H.); fanwei_mbb@126.com (W.F.); wangquan03@caas.cn (Q.W.); 2Academy for Advanced Interdisciplinary Studies, Nanjing Agricultural University, Nanjing 210095, China; lifeever@163.com; 3Guangdong Provincial Key Laboratory for Plant Epigenetics, Longhua Bioindustry and Innovation Research Institute, College of Life Sciences and Oceanography, Shenzhen University, Shenzhen 518060, China; 18374993460@163.com

**Keywords:** translational regulation, Ribo-seq, ribosome pausing, maize, translatomics

## Abstract

Regulation of translation is a crucial step in gene expression. Developmental signals and environmental stimuli dynamically regulate translation via upstream small open reading frames (uORFs) and ribosome pausing. Recent studies have revealed many plant genes that are specifically regulated by uORF translation following changes in growth conditions, but ribosome-pausing events are less well understood. In this study, we performed ribosome profiling (Ribo-seq) of etiolated maize (*Zea mays*) seedlings exposed to light for different durations, revealing hundreds of genes specifically regulated at the translation level during the early period of light exposure. We identified over 400 ribosome-pausing events in the dark that were rapidly released after illumination. These results suggested that ribosome pausing negatively regulates translation from specific genes, a conclusion that was supported by a non-targeted proteomics analysis. Importantly, we identified a conserved nucleotide motif downstream of the pausing sites. Our results elucidate the role of ribosome pausing in the control of gene expression in plants; the identification of the *cis*-element at the pausing sites provides insight into the mechanisms behind translation regulation and potential targets for artificial control of plant translation.

## 1. Introduction

Modulation of translation is a well-conserved mechanism that regulates gene expression in numerous species, from bacteria to animals and plants [1]. In mammalian cells, translation is modulated to fine-tune embryonic development, cell differentiation, and the response to nutrient scarcity [2,3]. In plants, translation can be quickly regulated by biotic and abiotic stresses, showing potential in crop breeding for improved disease resistance [4]. Over the past decades, the general mechanisms of translation initiation, elongation, termination, and recycling have become well understood. Recent studies have revealed that the translation efficiency of individual transcripts is differentially regulated under various conditions. How the translation machinery recognizes various transcripts and adjusts the moving pace of translating ribosomes remains elusive.

The development of ribosome profiling (Ribo-seq), based on the high-throughput sequencing of mRNA fragments protected by bound ribosomes following the digestion of polysomes with RNaseI, has created avenues for the monitoring of translation efficiency for individual transcripts at single-codon resolution [5,6,7,8]. Analysis of ribosome occupancy on actively translated transcripts has identified many genomic loci that were previously considered noncoding regions, including 5′ untranslated regions (UTRs) and long noncoding RNA (lncRNA) [9]. In yeast (*Saccharomyces cerevisiae*) and mammalian cells, Ribo-seq data revealed many upstream open reading frames (uORFs) located in the 5′ end of the main open reading frame (mORF) encoded by a given transcript; these uORFs repress mORF translation [10,11]. Accumulating evidence from Ribo-seq and biochemical experiments has demonstrated that uORFs inhibit the translation of their downstream mORFs by several mechanisms, including depletion of limited transfer RNAs (tRNAs), disassociation of ribosomes from RNAs, and triggering of nonsense-mediated RNA decay (NMD) [12,13,14]. Comparative analyses between mice (*Mus musculus*), humans (*Homo sapiens*), and zebrafish (*Danio rerio*) revealed that uORFs are depleted from the regions near coding sequences (CDSs) and that the repression of their cognate mORFs is conserved across these species [15]. Although it was known that ribosomes do not always move along mRNAs at a constant rate, the precise position(s) of ribosome pausing became accessible only with the implementation of Ribo-seq.

Accumulating lines of evidence indicate that ribosome pausing at a given site on a transcript prolongs the probability for nascent peptides to interact with protein partners, facilitating co-translational folding, complex assembly, organelle targeting, and chaperone recruitment. In addition, ribosome pausing can be induced by environmental conditions such as nutrition scarcity and heat stress. For example, arginine limitation in mammalian cells led to ribosome pausing and lower protein biosynthesis rates [16]. Aging also induced ribosome pausing in the nematode *Caenorhabditis elegans* and budding yeast, resulting in an overload of the ribosome-related quality control system and the aggregation of nascent peptides [17].

More factors influencing ribosome pausing are being identified, such as the secondary structures of mRNAs, contiguous proline residues in nascent peptides, and aggregation of positive charges at peptide chain exits; notably, codon usage appears to have limited influence [18]. Eukaryotic initiation factor 5A (eIF5A), a ribosomal pause relief factor, was shown to regulate the selection of the translation initiation codon and maintain the fidelity of translation initiation in humans and yeast [19]. Another study unveiled the connection between RNA methylation and translation: ribosome pausing was detected at the initiation codon, which acted like a “brake” to restrict protein output. This mechanism was regulated by N⁶-methyladenosine (m6A) methylation modification near the initiation codon [20].

The mechanisms of translational regulation in plants are now better understood, thanks in large part to the Ribo-seq technique. Numerous genes were shown to be under translational modulation to promote photomorphogenesis and chloroplast development when plants become exposed to light [21,22,23]. Translational regulation is also involved in plant acclimation to environmental stresses. Indeed, drought, hypoxia, salinity stress, and nitrogen depletion also induce translational regulation in plants, with thousands of uORFs in transcripts bound by ribosomes in different plant species [24,25,26,27].

In Arabidopsis (*Arabidopsis thaliana*), uORFs upstream of *TL1-BINDING FACTOR 1* (*TBF1*), a key gene encoding an immunity-related transcription factor, repress TBF1 translation and play a vital role in resistance against pathogenic bacteria. Surprisingly, transformation of rice (*Oryza sativa*) plants with a *uORF_TBF1_-AtNPR1* construct comprising *uORF_TBF1_* placed upstream of the coding sequence for the immunity gene *NONEXPRESSER OF PR GENES 1* (*AtNPR1*) significantly enhanced blight resistance of rice without growth penalty, suggesting conservation of uORF-mediated translation regulation [28,29]. Further studies demonstrated that hairpin structures downstream of the uORF were required to suppress the translation of the mORF upon bacterial invasion [30].

Although recent progress indicates that translational regulation is a general and important mechanism by which plants respond to various developmental and environmental signals, whether plants undergo ribosome pausing is currently poorly understood. This study aims to reveal the genes under the regulation of ribosome pausing and identify the sequence features triggering the ribosome pausing in plants.

Light is not only the most important energy source for plant growth and development by powering photosynthesis, but it also provides essential signals to shape plants as they grow. Photomorphogenesis is the developmental program induced in dark-grown (etiolated) seedlings when they first perceive light. Previous studies in Arabidopsis and maize (*Zea mays*) seedlings have suggested the participation of translational regulation in plant responses to light, although no evidence for ribosome pausing has been reported [21,22,23]. Here, we exposed etiolated maize seedlings to light and performed transcriptome deep sequencing (RNA-seq) and Ribo-seq analyses at several time points. We identified numerous ribosome-pausing events in maize seedlings establishing photomorphogenesis that negatively regulated the translation efficiency of their cognate proteins. Furthermore, the conserved cis-element associated with ribosome pausing was revealed. Our finding provided global evidence that plant genes were broadly regulated by ribosome pausing, and the identified motif triggering the ribosome pausing could be utilized in artificial modulation of translation for plant transcripts and the molecular breeding of crops.

## 2. Results

### 2.1. Translational Regulation Responds Quickly to Early Light Exposure

We exposed 6-day-old etiolated maize seedlings from the inbred line B73 to various durations of white light (330 μmol/m^2^/s) and collected samples immediately before transfer to light (0 h time point) and after 0.5, 1, 2, and 4 h of light treatment for the generation of matched Ribo-seq and RNA-seq libraries. For each Ribo-seq library, we obtained over 12 million raw reads (Appendix A). From these, we removed ribosomal RNAs (rRNAs) with specific probes, retaining 6% to 30% of the initial raw reads (Appendix A). We mapped the remaining reads to the maize reference genome, which revealed lengths for most ribosome-protected fragments (RPFs) as 27 nucleotides (nt) to 31 nt (Appendix A), a range consistent with previous reports [8,23,24]. Over 49% of the identified RPFs were generated from annotated protein-coding genes, and more than 50% of the RPFs were located in short open reading frames (sORFs), including independent uORFs, downstream open reading frames (dORFs), and some uORFs overlapping with mORFs (Figure 1A).

Taking 29 nt reads at the 2 h time point as an example, the RPFs showed a clear 3 nt periodicity along the CDS, with the first nucleotide of most RPFs mapping to the +1 position of each codon (Figure 1B and Appendix A). Other libraries showed a similar periodicity (Appendix A). Along mRNAs, the RPFs accumulated about 13 nt upstream of the start codon and about 16 nt upstream of the stop codon, corresponding to the positions where ribosomes assembled and dissociated from the transcript, respectively. The nucleotide periodicity and the distribution pattern of the RPFs reflect the typical features of Ribo-seq data.

To assess the dynamics of translational regulation in response to light, we looked for differentially translated genes (DTGs) from four pairwise comparisons (0–0.5 h, 0.5–1 h, 1–2 h, 2–4 h, and 0.5–4 h). We discovered that translational regulation mainly occurs during the 0–0.5 h period, as the number of DTGs dramatically decreased for the later comparisons (Figure 2A–E). The shift from darkness to light is an abrupt environmental change for seedlings; the large number of DTGs at the earliest time point reflected the prompt responses of translational regulation to cope with sharp environmental fluctuations. This observation is in line with the idea that translational regulation is fast and can respond to environmental signals quickly [31]. Although the pairwise comparisons between 0.5 and 1 h, 1 and 2 h, and 2 and 4 h periods yielded no DTGs, the comparison of 0.5–4 h resulted in many DTGs, which may be attributed to a stable environment that led to a slow accumulation of gene expression changes (Figure 2E).

To assess the consistency in pattern between the changes in transcript levels and translation, we compared the differentially expressed genes (DEGs) and DTGs at various time points into light exposure (Figure 2F). From 0 to 0.5 h, we identified 1800 upregulated DEGs and 159 upregulated DTGs, with 68 shared genes. We also identified 1071 downregulated DEGs and 116 downregulated DTGs, with 35 shared genes. These shared genes accounted for about 3% of all DEGs but over 30% of DTGs. For the pairwise comparison of 0.5–4 h, we obtained 4977 upregulated DEGs and 137 upregulated DTGS, of which 115 were shared; likewise, we identified 3383 downregulated DEGs and 89 downregulated DTGs, of which 73 genes were shared. For this longer period, the shared genes accounted for about 2% of all DEGs and about 80% of all DTGs. Thus, the vast majority of changes in transcript levels are not accompanied by changes in translation, especially over short time periods. In addition, most changes in transcript and translation levels were in the same direction. Our results indicate that the consistency between transcript levels and translation dynamically changes. In the earliest stage of light exposure (0–0.5 h), the genes regulated in the same direction for their translational output and their transcript levels represent fewer than 43%, while translational regulation is highly consistent with transcript abundance in the later stage (0.5–4 h) (>82%).

### 2.2. Genes Are Differentially Regulated at the Transcript and Translation Levels during Photomorphogenesis

The poor apparent correlation between transcript levels and their translational level prompted us to check individual genes with different regulatory patterns at the two levels (Figure 3A–D). In the first 0.5 h of light treatment, most of the genes encoding photomorphogenesis regulatory factors remained unchanged either transcriptionally or translationally, except for *ELONGATED HYPOCOTYL5* (*HY5*) and its homolog *Zm00001d039658*, which were upregulated at both levels (Figure 3A). Among the negative regulators of photomorphogenesis, *CONSTITUTIVE PHOTOMORPHOGENIC1* (*COP1*), *FUSCA5* (*FUS5*), *B-BOX19* (*BBX19*), and *LIGHT-RESPONSE BTB1* (*LRB1*) were upregulated at the translational level, while their transcript levels were constant. *BBX20*, another negative regulator of photomorphogenesis, was downregulated at the translational level but unchanged at the transcript level (Figure 3A). Most of the transcripts from chloroplast genes were unchanged at the translational level, while the transcript levels of some chloroplast genes encoding ribosomal proteins, including *rpl14*, *rpl20*, *rpl32*, *rpl33*, and *rpl36*, were decreased, while those of *psbC*, *psbL*, *psaC*, *psbF*, *psbT*, *psbN*, and *rpl22* were increased (Figure 3A).

From 0.5 to 1 h, most photomorphogenesis-related transcripts and chloroplast transcripts remained unchanged at both the transcript and translation levels (Figure 3B). During the period from 1 to 2 h, *PHYTOCHROME-INTERACTING FACTOR5* (*PIF5*) was specifically downregulated at the transcript level, while *FAR-RED INSENSITIVE219* (*FIN29*) was specifically upregulated at the translational level. Some chloroplast transcripts encoding chloroplast ribosome proteins and photosystem components were significantly upregulated at the transcript level but not at the translational level (Figure 3C). With prolonged light exposure (more than 2 h), some chloroplast transcripts were upregulated at the translational level with mild changes in their transcript levels (Figure 3D). Notably, *HY5* transcript levels were lower, but the translation level remained unchanged.

We calculated the translation efficiency (TE) of each transcript to evaluate its utilization by ribosomes [32]. In general, the TE of most transcripts, including key regulators of photomorphogenesis, did not change much, remaining below the significance threshold set for this study (|Z-score| > 1.5). However, several other regulators of photomorphogenesis were affected by exposure to light. For example, the TE of *FUS5* was slightly enhanced at 0.5 h compared to the dark control (Figure 3E), and the TE of *COP1* moderately decreased under 0.5 to 1 h of light exposure (Figure 3F). The TE of *EIN3-BINDING F BOX PROTEIN* (*EBF*) and *HY5* increased significantly between 1 and 2 h and after 2 h, respectively (Figure 3G,H). Following 0.5 h of light exposure, the TE of the chloroplast genes *psaC*, *rpl22*, *rps14*, and *psbF* were higher compared to their TE in darkness. From 1 to 2 h of light exposure, some photosystem genes showed lower TEs (Figure 3G), while the transcripts encoding chloroplast ribosome proteins and electron transfer chain components exhibited the opposite pattern with higher TEs (Figure 3H).

To investigate the correlation between the changes in transcript levels and TE, we plotted the fold-changes in RNA abundance and the corresponding TE for each gene. In general, the changes in TE and transcript levels were negatively correlated, especially in the later periods, as might have been expected, with Pearson’s correlation coefficients (R) of R_0–0.5 h_ = −0.38, R_0.5–1 h_ =−0.65, R_1–2 h_ = −0.70, and R_2–4 h_ = −0.76 (Figure 3I–L). *HY5* always maintained efficient translation independently of changes in its transcript levels, suggesting the specific regulation of its TE by light. By contrast, *COP1* showed an increase in TE and transcript levels from 0 to 0.5 h, followed by a lower TE but a higher RNA abundance from 0.5 to 1 h.

### 2.3. Light Exposure Widely Alleviates Ribosome Pausing in Maize

The inconsistent patterns observed between translation and transcript levels suggested the existence of translational regulation. Ribosome pausing, an important translational regulatory mechanism, has been widely studied in prokaryotes. To investigate whether ribosome pausing participated in photomorphogenesis, we calculated the pausing score for 5,115,551 ribosome-protected sites and identified 466 paused transcripts (pausing score > 50, Z-score > 1.65, and base count > 20 as a cutoff) across five time points (Figure 4A). Overall, most ribosome-pausing events were specific to etiolated seedlings but were quickly released after a short exposure to light (0.5 h). We divided the transcripts undergoing ribosome pausing into five clusters according to their extent of ribosome pausing at each time point (Figure 4B and Appendix A, Appendix A). Cluster 1 exhibited a significant increase in ribosome pausing at 0.5 h, returning to low levels at later time points. In Clusters 2 and 3, the ribosome-pausing levels quickly decreased following exposure to light; in Clusters 4 and 5, the degree of ribosome pausing first dropped sharply before raising again during one of the later time points.

To understand the functions of the transcripts with ribosome pausing, we performed a Kyoto Encyclopedia of Genes and Genomes (KEGG) pathway and Gene Ontology (GO) term enrichment analysis using the genes from each cluster (Figure 4C, Appendix A, Appendix A). Cluster 1 genes were enriched for ‘macromolecule modification’ and ‘regulation of developmental processes’. *Zm00001d039637* encodes Golden2-like14 (GLK14), and its homolog EARLY FLOWERING MYB PROTEIN (EFM) in Arabidopsis interacts with a negative regulator of H3K36 methylation [33]. Light is an indispensable condition for plant growth, but sudden exposure to light can also cause physiological stress to the etiolated seedlings. *Zm00001d023240* encodes the putative Ser/Thr protein kinase BLUE LIGHT SIGNALING1 (BLUS1), which mediates an early step in phototropism signaling in guard cells under light conditions. Various stresses inhibit *BLUS1* expression through the production of reactive oxygen species (ROS), thereby closing stomata [34]. *ALTERNATIVE OXIDASE1a* (*AOX1a*) and the cytochrome P450 gene *CYP81D8* encode proteins responsible for ROS clearance, and their expression is induced by elevated ROS levels. The transcript levels of these two genes were upregulated at the 0.5 h time point relative to the 0 h time point (darkness), indicative of ROS accumulation in plant cells (Figure 4C and Appendix A). The presence of *BLUS1* in Cluster 1 suggests that ribosome pausing is involved in acclimation to light stress.

Transcripts in Clusters 2 and 3 were involved in the ‘regulation of seedling development’, the ‘misfolded protein response’, the ‘hydrogen peroxide metabolic and catabolic process’, ‘protein dephosphorylation’, and ‘tRNA maturation’. *Zm00001d022045* encodes tRNA (guanosine (18)-2’-O) methyltransferase and is predicted to localize to chloroplasts. The release of ribosome pausing on this transcript in the light would be beneficial for tRNA maturation and chloroplast development. *Zm00001d002131* encodes STRESS-ENHANCED PROTEIN2 (SEP2), a chlorophyll-binding protein. In Arabidopsis, the homolog *AtSEP2* was transcribed under low-light conditions and rapidly induced by high-light conditions. Our RNA-seq data confirm this conclusion (1.87-fold increase, Appendix A). The ribosome pausing of this gene was gradually released after 0.5 h, indicating that its light-responsive expression is regulated at the transcript and translation levels. The functions of genes in Clusters 4 and 5 were mainly involved in the ‘microtubule-based process’ and ‘RNA 3′ uridylation’. *Zm00001d015059* encodes UTP: RNA uridyltransferase 1 (UTR1), the main terminal uridyltransferase (TUTase) responsible for mRNA uridylation. UTR1 can repair de-adenylated mRNA ends and maintain mRNA stability [35]. These results show that exposure of etiolated seedlings to light can release ribosome pausing, which may be an important mechanism for the regulation of gene expression to cope with abrupt environmental changes.

We looked specifically for all important regulatory factors of photomorphogenesis and chloroplast transcripts for ribosome-pausing events. However, their respective transcripts were not included in the list of transcripts experiencing significant and high ribosome pausing, suggesting that ribosome pausing is not the main translational regulatory mode of these transcripts, despite some individual chloroplast transcripts displaying some pausing (Appendix A). Some mitochondrion transcripts, by contrast, displayed changes in ribosome pausing (Appendix A, Appendix A).

### 2.4. Ribosome Pausing Negatively Regulates Translation in Maize

Based on reports from bacteria, yeast, Arabidopsis, and cancer cells, paused ribosomes negatively regulate protein biosynthesis [36,37,38,39]. To examine the effects of ribosome pausing on plant translation, we characterized the TE changes for the genes from each of the five clusters in the 0–0.5 h pairwise comparison (Appendix A). Notably, most of these paused transcripts did not show significant changes in their TE for this time interval. In Cluster 1, the TE of 15 and 2 transcripts was significantly upregulated and downregulated, respectively. In Clusters 2–5, very few transcripts exhibited an increased TE, with relatively more transcripts experiencing a decrease in their TE during the 0–0.5 h interval.

We did not find any relationship between TE and pausing score (Figure 5A), which prompted us to reassess our analysis method to calculate TE. Accordingly, we randomly chose several paused transcripts; their degree of pausing was always negatively related to the extent of RPF coverage (measured as the ratio of nucleotides in RFPs to those in the CDS) (Figure 5B and Appendix A). Usually, TE uses the read counts from Ribo-seq and RNA-seq data for single transcripts and ignores ribosome coverage. When transcripts undergo ribosome pausing, the pausing sites will be highly represented among the reads in the Ribo-seq data, whereas the rest of the sites in the transcript will not be captured. Therefore, unevenly distributed RPFs along a transcript undergoing pausing events may have an unchanged or even higher TE. To solve this issue, we calculated the translation intensity (TI, TI_transcript_ = Median_RPFs_·Coverage_RPFs_), which takes into account read coverage, to authentically reflect translation progression. We found the pausing score and TI usually changed in the reverse direction. Taking the 2 h and 0 h time points as an example, we observed that genes with higher pausing scores at 0 h have a lower TI compared to that at 2 h (Figure 5C). We conclude that ribosome pausing negatively regulates translation in plant cells.

To further test for negative regulation of pausing on translation, we performed a tandem mass tag labeling quantitative proteomics analysis of samples from the 0 h and 2 h time points. The proteomics data indicated that the genes whose transcripts are characterized by higher pausing scores produce less protein (Figure 5D and Appendix A). We chose transcripts whose pausing scores decreased by over 100 at the 2 h time point relative to the 0 h time point and individually inspected the quantification data from the proteomics analysis. A total of 14 proteins increased their abundance at 2 h compared to 0 h among 17 significantly changed proteins (Appendix A). In summary, ribosome pausing plays a negative regulatory role in protein biosynthesis in maize.

### 2.5. A High GC Content Leads to Ribosome Pausing

As an important regulation of protein translation, ribosome pausing occurs at specific positions along certain transcripts. To reveal the possible position preference of ribosome pausing, we calculated the distribution of pausing sites along the scaled length of transcripts, which revealed that most pausing sites were localized in the CDS, with few pausing events present in the 5′ UTR or 3′ UTR. In the CDS, pausing did not display a clear distribution bias, indicating that ribosome pausing might inhibit translation at the elongation step (Figure 6A).

In chloroplasts, the secondary structure of mRNA, contiguous proline residues in the encoded proteins, and positive charges carried by newly formed peptides are all associated with ribosome pausing [18]. In the study of bacteria, several factors affected the movement speed of ribosomes, such as Shine–Dalgarno sequences, pro-pro motifs, and charged amino acids mediating specific interactions between the ribosomal exit tunnel and the nascent peptide [37,40,41,42]. To investigate the sequence features linked to ribosome pausing in maize, we analyzed the encoded amino acids around the pausing sites, the charges of their nascent peptide chains, and the sequence conservation of the transcripts. We did not identify clear pro-pro motif characteristics in the nascent peptide sequences around the pausing sites or a clear pattern in their charge properties (Appendix A). We thus looked for a sequence feature in the transcripts undergoing ribosome pausing. Importantly, we detected a clear C signal at 13 nt upstream of the pause sites (Figure 6B). This signal should not be caused by RNase I digestion, as RNase I has no base preference [43]. We also checked the sequences of random RPFs on transcripts without ribosome pausing and failed to find this C-rich motif, suggesting that the upstream C-rich feature is specific for transcripts subjected to ribosome pausing (Figure 6B). In addition, we compared the GC content between pausing and non-pausing transcripts using 500 bp of sequences on either side of the pausing sites or of random sites from non-pausing RPFs. The GC ratio around the pausing site was higher than that of other ribosome-protected sites not undergoing pausing (Figure 6C).

To investigate whether pausing genes and other genes have different GC biases, we inspected the GC content of full-length transcripts centered on the RPF sites and scaled the length of the transcripts from the start codon to the pausing site and from the pausing site to the stop codon into 50 bins, respectively. When we aligned the start codon, stop codon, and the pausing site or ribosome-protected site from pausing and non-pausing transcripts, we discovered that the GC content of full-length pausing transcripts is higher than that of the non-pausing transcripts over the entire transcript length (Figure 6D). This result suggests that the GC content distribution of entire transcripts rather than that of a fragment of the transcript is related to ribosome pausing. Most transcripts show a gradual decrease in their GC content from the start codon to the stop codon [44]. Although we observed this trend for pausing transcripts and the random set of transcripts, the GC content near the pausing site increased more than at random ribosome-protected sites. To our knowledge, in previous studies on ribosome pausing in bacteria, yeast, and animal cells, no relationship has been described between the GC content of transcripts and ribosome pausing, which may be a unique mechanism for ribosome pausing in plants.

We used MEME tools to search for potential conserved motifs in the regions upstream and downstream of pausing sites separately, identifying the significant motif “CGCCGCCGCCGCCGCC” (CGC motif) in the 3′ region of the pausing sites (Figure 7A and Appendix A). The transcripts containing this CGC motif had a significantly higher pausing score than transcripts lacking this motif (Figure 7B). The free energy of the thermodynamic ensemble of the CGC motif is −2.95 kcal/mol, which facilitates the formation of stem-loop structures based on simulations (Figure 7C). We analyzed the functions of all transcripts with a CGC motif and those experiencing ribosome pausing in our Ribo-seq data. These genes were involved in many important pathways, including signaling transduction, molecular interaction, and transcription (Figure 7D). Our results reveal sequence features that might be previously unknown determinants of ribosome pausing in maize.

## 3. Discussion

Translational regulation plays pivotal roles when plants are faced with severe environmental changes, but the underlying mechanisms are not clear. In this study, we discovered that maize genes are specifically modulated at the translational level when etiolated seedlings are exposed to light, with a rapid alleviation of translation repression imposed by ribosome pausing. Furthermore, we determined that an upstream cytosine and a downstream CGC-rich motif were conserved features at the pausing sites. These results shed new light on the mechanisms behind translational regulation and provide a potential *cis*-element as a target for translational control of plant genes.

Translational regulation mainly responds to light during the initial period of light exposure. When seedlings or plants are faced with a marked environmental change, they must react quickly by controlling signal transduction and metabolism in their cells. Light is not only indispensable for plant growth as an energy source, but it is also a major environmental stress for etiolated seedlings. Our results show that translational regulation mainly occurs within 30 min after etiolated maize is exposed to light. For longer light durations, far fewer genes were regulated at the translational level, indicating prompt translational modulation in response to light in maize. Protein translation may constitute an ideal regulatory node for environmental acclimation in plants, as it can quickly promote or attenuate protein production by mobilizing existing RNAs or stopping their translation in cells without de novo transcription or mRNA degradation. Notably, several important regulators of photomorphogenesis were upregulated at the translational level. COP1 is a master regulator of photomorphogenesis by inducing the degradation of key transcription factors such as HY5 and BBX in the light [45,46]. We found that *COP1*, *HY5*, and two *BBX* genes were all under translational modulation when etiolated maize seedlings were exposed to light, indicating that protein homeostasis is important for photomorphogenesis and is fine-tuned not only by protein degradation but also by translation.

Ribosome pausing is widespread in plants as a mechanism to repress the translation of individual transcripts. We identified 466 ribosome-pausing events with altered pausing scores among the different time points of light exposure. Consistent with their regulation at the translation level, we detected most of the ribosome-pausing events in etiolated seedlings before light exposure, suggesting a strong relationship between ribosome pausing and darkness. Following the initial transfer into the light, most of the transcripts experiencing ribosome pausing in the dark were released from this repression, leading to global translational changes (0–0.5 h). The pausing scores of individual transcripts increased or decreased during the 0–0.5 h interval. Transcripts for genes involved in macromolecular modifications, such as those encoding protein kinases, phosphatases, and ubiquitin ligases, had lower pausing scores in darkness, suggesting modulation of protein homeostasis and signaling pathways by ribosome pausing in darkness. In addition to providing an energy source and a photomorphogenesis signal, a sudden light illuminating etiolated seedlings also causes physiological stress. Genes involved in stress were also found to be differentially regulated by ribosome pausing. The increased ribosome pausing along *BLUS1* transcripts and the decreased ribosome pausing along transcripts of genes involved in the misfolded protein response indicate that light causes severe stress to etiolated seedlings, with an important role for ribosome pausing in alleviating this stress.

Previous studies revealed that ribosome pausing repressed translation progression [2,16,36,37,38,39,47,48,49,50]. Our results agree with this conclusion based on two lines of evidence. First, we observed a scarcity of RPFs around the paused site, which itself is enriched in RPFs. As an unintended consequence, fewer RPFs on both sides of the ribosome-pausing site, together with the high number of RPFs at the paused site, might mask changes in TE compared to random sites with no ribosome pausing. Indeed, the RPF coverage across the CDS was generally lower when ribosome pausing occurred, which prompted us to introduce the parameter TI to reflect true translation activity. We detected a clear negative relationship between ribosome pausing (measured as a pausing score) and translation activity (as estimated by TI). Second, we performed quantitative proteomics analysis of etiolated seedlings (0 h time point) and after 2 h of light exposure, which confirmed the negative relationship between ribosome pausing and protein production.

The secondary structure of mRNA is crucial to causing ribosome pausing. Our study found a significant downstream CGC motif whose presence was strongly correlated with ribosome pausing. Considering the large number of secondary structures in mRNAs, it is unlikely that multiple secondary RNA structures can efficiently impede translation. Several studies have demonstrated that stem-loop structures can slow down translation elongation [18,51,52,53,54]. The length of RNA stem-loops was also reported to be crucial for translational repression [55]. Additionally, evidence from Arabidopsis indicated that downstream hairpins can affect the selection of the upstream start codon (uAUG), with the bacterial elicitin elf18 inducing RNA helicases to rescue translation inhibition by loosening hairpins [30]. Although secondary structures were known to induce ribosome pausing, no specific sequence had been identified. Our study discovered a significant motif that is important for ribosome pausing and translational inhibition through the formation of a stem-loop structure. Although its function in ribosome pausing still needs to be verified by experimental investigations, the motif identified in this study provides potential *cis*-element targets for engineered translational regulation in plants.

## 4. Materials and Methods

### 4.1. Maize Seedling Growth and Treatments

Seeds for the maize (*Zea mays*) inbred line B73 were surface sterilized with 3% (*w*/*v*) sodium hypochlorite for 10 min before being washed with distilled water five times. The disinfected seeds were sown in pots containing dump soil (0–6 mm, PINDSTRUP) and grown in a growth chamber at 27 °C with a humidity of 65% in the dark for 6 days. After 6 days of darkness, the leaves of the etiolated seedlings were cut and placed in liquid nitrogen as the 0 h samples. The remaining seedlings were exposed to light by transferring the pots to a plant culture shelf at 27 °C under a white light intensity of 330 μmol/m^2^/s; leaves were collected at 0.5, 1, 2, and 4 h into light exposure. We took this culture process 3 individual times to reduce operational errors.

### 4.2. Ribo-Seq and RNA-Seq Library Construction

A previously described method was modified to isolate ribosome-protected fragments (RPFs) [8]. In detail, 0.5 g (fresh weight) of leaf tissues was ground in liquid nitrogen to a fine powder, to which 2.5 mL polysome extraction buffer (150 mM Tris-HCl, pH 8.0, 40 mM KCl, 20 mM MgCl_2_, 2% [*v*/*v*] polyoxyethylene, 0.4% [*w*/*v*] sodium deoxycholate, 1.5 mM dithiothreitol, 50 μg/mL chloramphenicol, 50 μg/mL cycloheximide, and 10 units/mL DNase I) was added and kept on ice for 15 min. The slurry was filtered through one layer of Miracloth and centrifuged at 16,000× *g* for 10 min at 4 °C. Total RNAs were isolated by Tri-reagent (Merck KGaA, Darmstadt, Germany) for RNA-seq. For Ribo-seq, 200 units of RNase I were added to 450 μL of supernatant (25 μg RNA) and incubated at 25 °C for 1 h. Ribosome monomers were isolated with size-exclusion columns (Illustra MicroSpin S-400 HR Columns; GE Healthcare, Chicago, IL, USA). Ribosome-protected RNA fragments were isolated with Tri-reagent. Ribo-seq libraries were constructed based on the described method with some modifications [56]. In brief, after preliminary screening by 10% PAGE with 7 M urea, 27 to 34 nt RPFs were recovered, and the phosphate group of the RNAs was treated with T4 PNK, followed by 3′ adaptor ligation (5′-rAppGATCGGAAGAGCACACGTCT–NH2) using truncated T4 RNA ligase 2 (NEB). Reverse transcription was performed with SuperScript II (ThermoFisher Scientific, Waltham, MA, USA) and an RT primer (5′-GATCGTCGGACTGTAGAACTCTGAACGTGTAGATCTCGGTGGTCGCCGTATCATT/iSp18/CACTCA/iSp18/CAGACGTGTGCTCTTCCGATCT). The reverse transcription was performed at 42 °C for 1.5 h, followed by incubation at 70 °C for 10 min. First-strand cDNA for ribosomal RNA (rDNA) was removed from the cDNA by hybridization with rDNA probes (Appendix A). All the probes and the primers used in following test were designed by our library and synthesized by Sangon (Shanghai, China). RPFs were circularized with Circligase ssDNA ligase (Epicentre CL4111K) and amplified by PCR (13 cycles, 60 °C annealing, and primer sequences are listed in Appendix A).

The Ribo-seq libraries were sequenced as paired-end 150 bp reads or single-end 75 bp reads on an Illumina NovaSeq 6000 instrument at Shanghai PersonalBio Technology (Shanghai, China). The RNA-seq libraries were sequenced on an Illumina NovaSeq 6000 instrument.

### 4.3. Quantitative PCR

First-strand cDNA was synthesized from total RNA samples using SuperScript IV (ThermoFisher scientific) and an oligodT primer. The reverse transcription was performed at 42 °C for 1.5 h, followed by incubation at 70 °C for 10 min. Quantitative PCR (qPCR) was performed with SYBR Green qPCR mixtures (ThermoFisher scientific, Waltham, USA) and gene-specific primers (Appendix A). *Ubiquitin* (*Ubi*, *Zm00001d053838*) was selected as an internal control.

### 4.4. TMT-Labeled Mass Spectrometry Analysis

Leaf samples (about 100 mg, fresh weight) were ground into a fine powder in liquid nitrogen and homogenized in 1 mL extraction buffer (0.9 M sucrose, 0.5 M Tris-HCl, 50 mM EDTA, 0.1 M KCl, 1% [*v*/*v*] Triton X-100, 2% [*v*/*v*] β-mercaptoethanol, and 1% [*w*/*v*] protease inhibitor cocktail set VI [Calbiochem], pH 8). To the above mixture, 1 mL of saturated phenol with Tris-HCl (pH 7.5) was added, and the upper phenolic phase was separated from the aqueous phase by centrifugation at 8000× *g* for 10 min at 4 °C. The upper phase was transferred to a fresh tube, to which five volumes of pre-cooled 0.1 M ammonium acetate in methanol were added and kept at −20 °C overnight. The proteins were pelleted by centrifugation at 10,000× *g* for 15 min at 4 °C; the pellet was washed with pre-cooled methanol and then acetone. Air-dried pellets were resuspended with 300 μL of lysate solution (6 M urea, 50 mM ammonium bicarbonate, pH 8.0) and incubated for 3 h at room temperature. Protein digestion, TMT labeling, and mass spectrometry were performed by Shanghai Luming Biological Technology (Shanghai, China). All analyses were performed with a Q Exactive HF mass spectrometer (ThermoFisher Scientific, Waltham, MA, USA) equipped with a Nanospray Flex source (ThermoFisher Scientific). Samples were loaded and separated by an Agilent Zorbax Extent C18 column (2.1 × 150 mm, 5 µm) on an Agilent 1100HPLC (ThermoFisher Scientific).

ProteomeDiscoverer 2.4 (ThermoFisher Scientific) was used to search the raw data thoroughly against the Uniprot taxonomy_4577 database. The alkylation of cysteine was considered a fixed modification during the search. For protein quantification, TMT was selected. The global false discovery rate (FDR) was set to 0.01, and protein groups considered for quantification required at least one peptide.

### 4.5. Sequencing Quality Control

Raw RNA-seq and Ribo-seq reads of low quality and adapter sequences were trimmed by fastp (v0.22.0) [57]. The clean resulting Ribo-seq reads were subjected to a length cutoff of 23–36 nt, with no undetected base allowed (“N” base number ≤ 0), a minimum quality score > 15, and paired reads correction. Raw RNA-seq reads were cleaned with default parameters. In total, ~81 M clean RNA-seq reads and ~30.5 M Ribo-seq reads were obtained from each replicate. After cleaning, paired reads from the Ribo-seq data were merged into a single read with the tool NGmerge (v0.2) [58] with parameters “-n 4 -p 0.05 -z -g.” FastQC (v0.11.9) [59] was used to analyze the quality of the reads, and MultiQC (v1.11) [60] was subsequently used for integrated comparative analysis of all samples.

### 4.6. Sequence Alignment to the Maize Reference Genome

The full sequence of the maize B73 genome and its annotation (v4.48) were downloaded from MaizeGDB [61]. Index files were created using three building methods for analysis. The three methods were: bowtie [62] (default parameters), bowtie2 [63] (default parameters), and STAR [64] (with parameters --runThreadN 8 --runMode genomeGenerate --genomeDir –genomeFastaFile --sjdbGTFfile). Before formal alignment, clean Ribo-seq reads were mapped to the rRNA and tRNA reference sequences for elimination by TopHat2 (v2.1.1, with bowtie1). Then, the clean remaining reads were aligned to the reference genome with gff annotation by TopHat2 (--segment-length 20 -N 2 -I 50000 --bowtie1 --segment-mismatches 2 -G RefGen_v4.48.gff3). For ORF and pausing analyses, the alignment was also performed using STAR (v2.7.9a) software against the entire genome, yielding two scales for aligned reads: at the single gene level and at the reference genome level. Alignment of the RNA-seq reads to the maize B73 reference genome was performed using TopHat2 (v2.1.1, with bowtie2) software. Then, samtools (v1.6) [65] software was used to sort and index the bam files.

### 4.7. Feature Read Counts and Differential Expression Analysis for RNA-Seq and Ribo-Seq Reads

For mapped RNA-seq reads, the union mode of HTSeq (v0.13.5) [66] software was used to obtain the number of reads mapping to different structures (exons and CDSs) based on each feature (gene, transcript). Differential expression analysis (filter: |log_2_(fold change)| > 1 and *p*-value < 0.05) and gene count normalization were processed by DESeq2 (v1.24.0) [67]. For Ribo-seq, featureCounts (v2.0.1) was used to calculate RPF counts. Following the identification of differentially abundant RPFs (filter: |log_2_(fold change)| > 1, *p*-value < 0.05, and no less than an average of five reads in support), analysis was also performed by DESeq2. For all samples, results were obtained as normalized counts, TPM, and FPKM values. MA plots were generated using the R package ggplot2 (v3.3.3) [68]. The translation efficiency (TE) level, TE value, and differential TE were calculated with the xtail package (v1.1.5) [32]. Ribosome pausing and uORF detection require transcription scale mapping results as an input. Thus, Ribo-seq data were mapped to the reference genome with STAR before merging the results from each replicate for each sample.

### 4.8. Trait Detection of Ribosome-Protected Fraction and uORF Prediction

The distribution of RPFs was plotted with the tool metaplot in the software RiboCode (v1.2.11) [69]. The transcription annotation gtf file was converted by the prepare_transcripts tool. Based on the distribution of read counts for each read length, the protection range and the P-site for different read lengths were obtained for uORF and pausing analyses. RiboCode main tool (parameters: “-l no -g -b”; read length: 24–36 nt) was applied to the following analysis, including uORF prediction. The bar plot and pie graph of distribution and trait statistics were also obtained with RiboCode.

### 4.9. Ribosome-Pausing Detection and Analysis

The pausing score and z-score of all pausing scores at the same coverage bins were calculated by the software Pausepred (v5.18.2) [70]. The screening cutoff was set with a pausing score > 50, a z-score > 1.65, and read counts in position > 20. To investigate the biased characters related to ribosome pausing, transcript sequences were aligned with the pausing site as the central position. Nucleotide and amino acid distributions were compared between genes whose transcripts showed pausing and 2000 randomly chosen transcripts. To investigate the GC content around the pausing site, we calculated the average GC content on two scales: ±500 absolute base position around the pausing site (RPF binding site at random) and cut each gene from the start codon to the pausing site (RPF binding site at random) and the pausing site to the termination codon into 50 relative bins separately. To identify potential motifs causing the pausing, 250 bp of sequence upstream and downstream (including the pausing site) were separately extracted and subjected to a motif scan. The motifs in ribosome-pausing regions (250 bp on either side of the pausing site) were analyzed with MEME suite (v5.5.2; parameters: -minw 3 -maxw 20 -nmotifs) [71]. The significant motifs were then converted to a position weight matrix (PWM) based on the MEME frequency results. To examine the relationship between pausing level and the presence of the motif, the longest transcripts of all genes with the PWM motif were scanned using the matrix-scan tool package in RSAT (parameters: -v 1 -pseudo 1 -decimals 1 -1str -origin end -bgfile longest.fa.8bp.freq -bg_pseudo 0.01 -return limits -return sites -return pval -return rank -return normw -return weight_limits -return bg_residues -lth score 1 -uth pval 1e-4-seq_format fasta -n score) [72]. The 8 bp base frequency table of all genes in the reference genome, which was used by the scan process as a background, was generated with the oligo-analysis tool in RSAT (parameter: “-l 8 -quick -v 1 -1str”). To determine the variation in pausing across time points, the maximum pausing score of all positions in the pausing transcript was calculated for each time point. The genes were then clustered based on the pausing score of their transcripts with the R package pheatmap [73].

### 4.10. Translation Intensity (TI) Calculation

The TI was calculated as follows:$${TI}_{transcript}={Median}_{RPFs}\cdot {Coverage}_{RPFs}$$
$$Coverage=\frac{n}{Length}$$
$$Median={Depth}_{base\left(\frac{n+1}{2}\right)}\cdot \left(n\%2\right)-\left[{Depth}_{base\left(\frac{n}{2}\right)}+{Depth}_{base\left(\frac{n}{2}+1\right)}\right]\cdot [\left(n\%2\right)-1]$$

*Median*: median depth of transcript bases covered by RPFs.

*Coverage*: transcript base coverage (%).

*n*: count of bases covered by RPFs.

*Length*: transcript length.

*Depth*: RPFs counts of base.

### 4.11. Gene Ontology and KEGG Enrichment Analysis

The transcripts exhibiting significant ribosome-pausing sites were turned into a gene list that was submitted to the Gene Ontology [74] website (https://geneontology.org/, accessed on 16 July 2023) for the analysis of biological processes, molecular function, and cellular components. For the KEGG enrichment analysis, the YuLab-SMU/createKEGGdb package and the command “createKEGGdb::create_kegg_db (‘zma’)” were used to download and construct the KEGG database. ClusterProfiler [75] was used for detecting enrichment. Both enrichment results are shown as bubble plots, drawn with ggplot2.

### 4.12. Graphical Visualization of Results

The distribution of read counts as a function of read length and sampling time point was plotted with the function geom_bar of the ggplot2 R package. Seqlogo was used to identify the conserved bases and amino acids of the protein being translated around the pausing site, using the ggplot2 ggpubr and ggseqlogo R packages. Pausing sites are locations of bases; amino acids are encoded by three bases; whatever the pausing site on 1, 2, and 3 of the codon, we set that amino acid as a pausing amino acid. The ORF is alone with the genome transcript annotation. Other regular dot plots, bar plots, box plots, and fold line plots were also drawn with ggplot2. Plots with significance markers were performed with the ggsignif package [76]. Plots with dot annotation texts were applied with the ggrepel package [77].

### 4.13. Secondary Structure Prediction and Free Energy Calculation

The prediction was performed by RNAfold (http://rna.tbi.univie.ac.at/cgi-bin/RNAWebSuite/RNAfold.cgi, accessed on 2 June 2023) with default parameters.

## Figures and Tables

**Figure 1 ijms-25-07985-f001:**
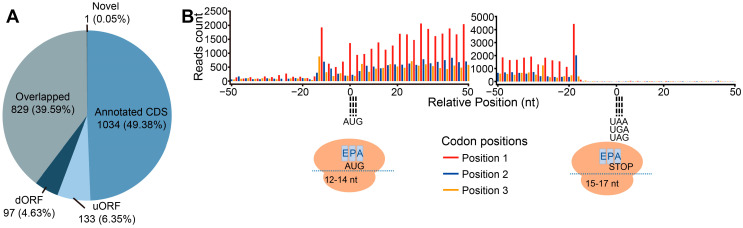
Genomic distribution, length, and 3 nt periodicity of identified ribosome-protected fragments (RPFs): (**A**) Distribution of RPFs across different features of the maize genome. Annotated coding sequences (CDSs), upstream open reading frame (uORF), downstream ORF (dORF), overlapped ORF, and novel coding regions are indicated with different colors. The numbers out and in parentheses indicate the gene number and the percentage of total reads, respectively. (**B**) Meta-gene analysis of the 29-nucleotide (nt) RPFs near the annotated translation start and stop sites in the maize genome. The red, blue, and orange bars represent the three possible open reading frames. E, P, and A indicate the aminoacyl-tRNA entry site, the P-site (peptidyl-tRNA formation site), and the E-site (uncharged tRNA exit site) in ribosomes, respectively. The numbers in the drawn ribosomes indicate the number of nucleotides protected by ribosomes upstream of the start codon and downstream of the stop codon.

**Figure 2 ijms-25-07985-f002:**
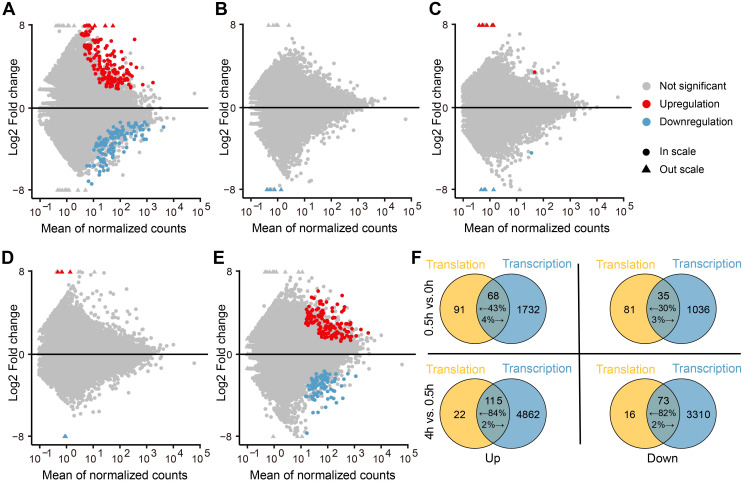
Translational regulation mainly occurs in the early stages of light exposure. (**A**–**E**) Differentially translational regulated genes (DTGs) following different durations of light exposure ((**A**), 0–0.5 h; (**B**), 0.5–1 h; (**C**), 1–2 h; (**D**), 2–4 h; (**E**), 0.5–4 h). Upregulated and downregulated DTGs are indicated with red and blue dots, respectively. (**F**) Venn diagrams showing the extent of overlap between differentially expressed genes (DEGs) and DTGs responsive to light exposure. DTGs and DEGs are indicated by the blue and yellow circles, respectively. The numbers represent the specific and common DEGs and DTGs.

**Figure 3 ijms-25-07985-f003:**
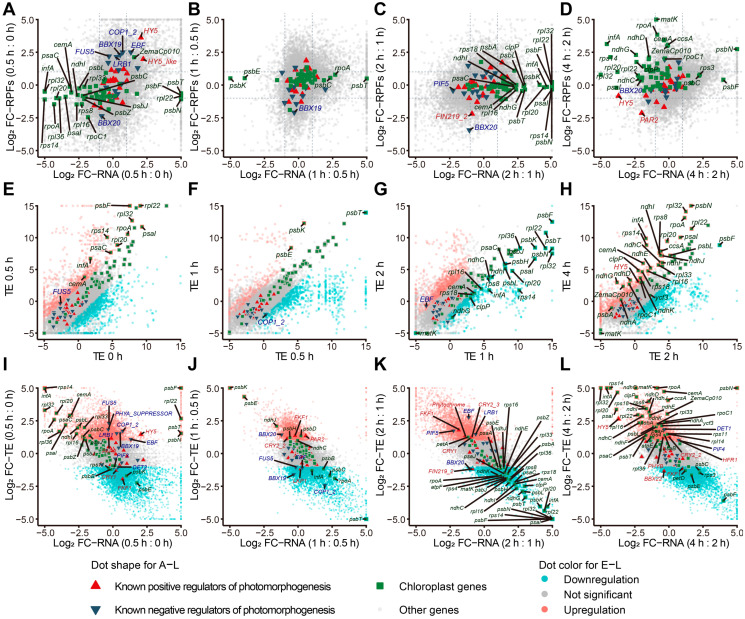
Expression patterns of photomorphogenesis-related transcripts at different time points: (**A**–**D**) Scatterplots of the fold-changes in RNA or RPF abundance following various durations of light exposure: (**A**), 0–0.5 h; (**B**), 0.5–1 h; (**C**), 1–2 h; (**D**), 2–4 h. All values are log2-normalized fold-changes between the listed time points. (**E**–**H**) Scatterplots of the translation efficiency (TE) following various durations of light exposure: (**E**), 0–0.5 h; (**F**), 0.5–1 h; (**G**), 1–2 h; (**H**), 2–4 h. (**I**–**L**) Scatterplots of log2-normalized fold-changes in TE or RPF abundance following various durations of light exposure: (**I**), 0–0.5 h; (**J**), 0.5–1 h; (**K**), 1–2 h; (**L**), 2–4 h. Positive and negative regulators of photomorphogenesis are indicated with red and blue triangles, respectively. Chloroplast transcripts are indicated in green rectangles. Upregulated, downregulated, and unchanged transcripts are marked with pink, cyan, and gray dots, respectively.

**Figure 4 ijms-25-07985-f004:**
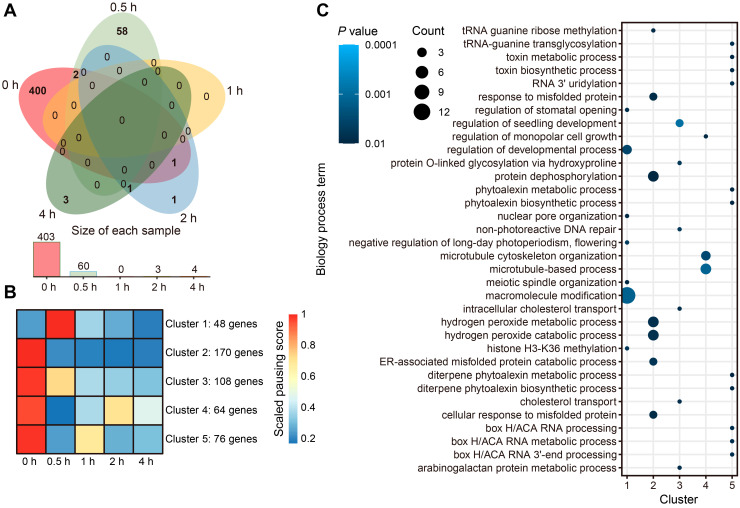
Genome-wide light-responsive ribosome-pausing events in maize-etiolated seedlings: (**A**) Venn diagram showing the extent of overlap between significant ribosome-pausing events identified at each of the time points of illumination. The numbers indicate pausing events specific to each sample or common to different samples. (**B**) Clustering analysis of transcripts showing ribosome-pausing events defining five clusters. The color scale indicates the strength of ribosome pausing. (**C**) Gene ontology (GO) term enrichment analysis (biology processes) of the genes whose transcripts belong to one of the five clusters defined above. The size of the circles indicates the number of genes; the color indicates the P value.

**Figure 5 ijms-25-07985-f005:**
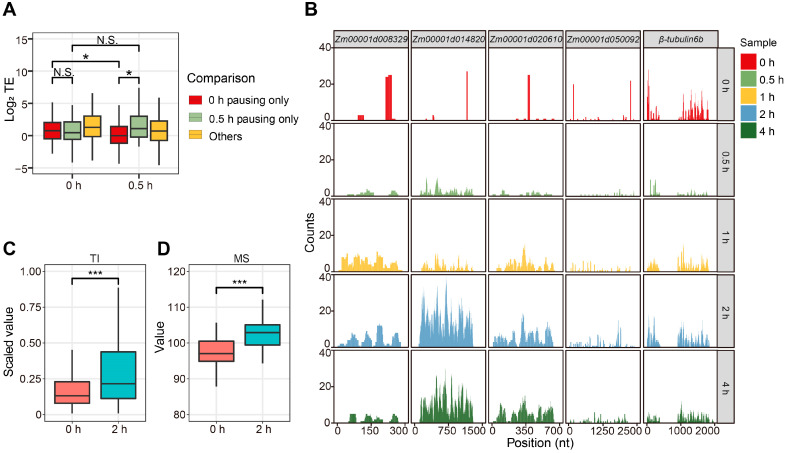
Ribosome pausing negatively regulates translation in maize. (**A**) The Wilcoxon test was used to assess significant differences in TE for transcripts showing ribosome pausing at different time points of light exposure. Significant mark * for *p* value < 0.05. (**B**) Distribution and coverage of RPFs along five randomly chosen transcripts showing ribosome pausing at different time points of light exposure. β-tubulin 6b is a non-pausing control. (**C**) Translation intensity (TI) for transcripts with high pausing scores at 0 or 2 h into light exposure. Significant mark *** for *p* value < 0.001. (**D**) Protein abundance, based on mass spectrometry analysis, translated from transcripts with high pausing scores at 0 or 2 h into light exposure. Significant mark *** for *p* value < 0.001.

**Figure 6 ijms-25-07985-f006:**
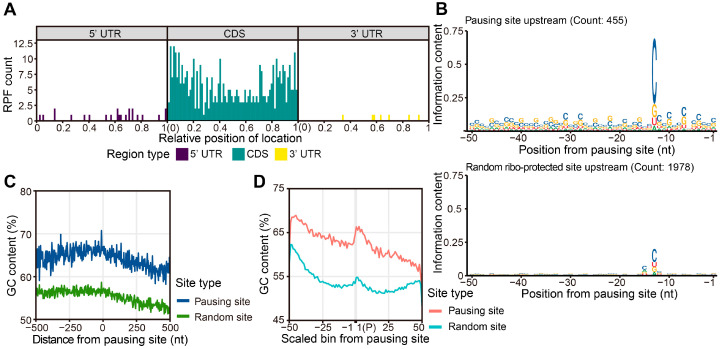
Sequence features of ribosome-pausing sites: (**A**) Meta-analysis showing the distribution of ribosome-pausing sites along different regions of maize transcripts. UTR, untranslated region. (**B**) Sequence logo of the region upstream of ribosome-pausing sites. The height of each letter indicates their probability at the corresponding positions. The positions along the *x*-axis are relative to the ribosome-pausing sites. The –1 position indicates the upstream 1 nt to the ribosome-pausing site. (**C**) Metaplot of GC content around ribosome-pausing sites (blue) and random RPFs (green). The GC content is over 500 bp on either side of the ribosome-pausing sites. (**D**) GC content over full-length transcripts with ribosome pausing and random transcripts. The lengths of transcripts have been normalized.

**Figure 7 ijms-25-07985-f007:**
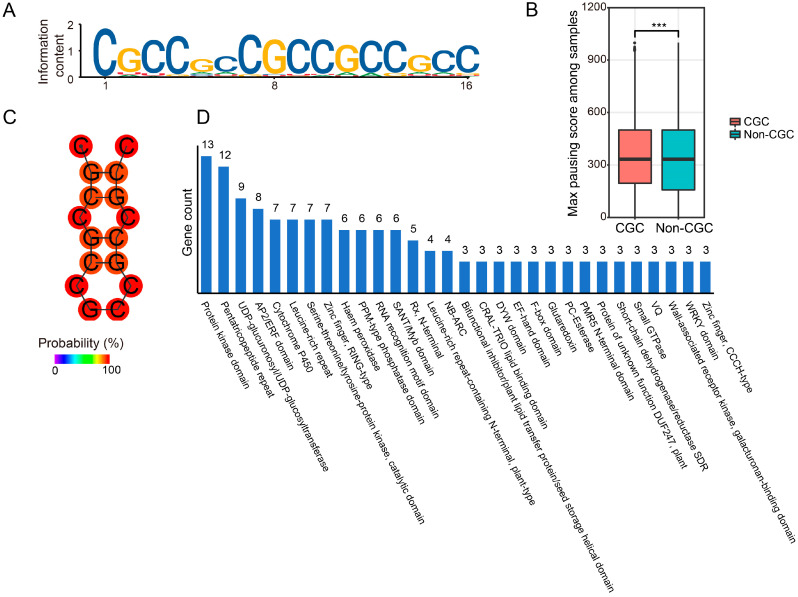
(**A**) A repeated CGC motif appears in the region downstream of ribosome-pausing sites. The height of each letter indicates the probability at the corresponding position. The positions along the *x*-axis are the length of the motif. (**B**) Maximum ribosome-pausing scores between transcripts with or without CGC motifs in all ribosome-paused transcripts. Significant mark *** for *p* value < 0.001. (**C**) Predicted secondary structure of the CGC motif. The possibilities of base-pairing are indicated as a gradient from blue (0%) to red (100%). (**D**) GO term enrichment analysis of genes whose transcripts contain the CGC motif.

## Data Availability

The data presented in this study are openly available from the National Genomics Data Center at https://ngdc.cncb.ac.cn, reference number PRlCA025750.

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
