# Peer review of "Ribosome Pausing Negatively Regulates Protein Translation in Maize Seedlings during Dark-to-Light Transitions"

_ijms, 2024, doi:10.3390/ijms25147985_

Round 1

Reviewer 1 Report

Comments and Suggestions for Authors

the authors used ribo-seq and bulk-RNA-seq to disclose the phenomenon of ribosome-pausing when maize seedlings exposed to light after growing in the dark, and found the negative correlation between ribosome pausing and translation efficiency. Overall, the authors presented lots of data and also found some meaningful correlation between light and ribosome pausing.  However, in the materials and methods, authors used the leaves without distinguish them, especially after light treatment which may hidden some useful information since leaves with different positions may have different physiological and biochemical reactions, may include translational efficiency. If authors used the fixed leaves for the comparison, maybe more meaningful data can be well collected. 

Author Response

Comments and Suggestions for Authors

the authors used ribo-seq and bulk-RNA-seq to disclose the phenomenon of ribosome-pausing when maize seedlings exposed to light after growing in the dark, and found the negative correlation between ribosome pausing and translation efficiency. Overall, the authors presented lots of data and also found some meaningful correlation between light and ribosome pausing.  However, in the materials and methods, authors used the leaves without distinguish them, especially after light treatment which may hidden some useful information since leaves with different positions may have different physiological and biochemical reactions, may include translational efficiency. If authors used the fixed leaves for the comparison, maybe more meaningful data can be well collected. 

Response:

We agree with the reviewer’s suggestion that a fixed parts of certain leaf may reveal more meaningful results. In this manuscript, we collected all leaves to construct the ribose based on the following reasons. First, the aims of this study are to reveal the translational regulation of maize seedlings facing the transition from dark to light. During this transition, light causes not only the photomorphogenesis but also stress to plant. Different leaves or leaf segments have various physiological responses mainly in terms of chloroplast development and photomorphogenesis, but the adaptative mechanisms to strong light stress may not have much differences between various leaves. Second, to ensure the etiolated status of the control plants, the materials of 0h samples must be collected in a period as short as possible. Separation of different leaves from seedling might take more time and cause weak responses to light for the control plants. Third, two literatures studying the translational regulation of light treatment on Arabidopsis collected the whole aerial parts of plants for the similar studies, and found wide-spread translational regulation events (https://doi.org/10.1105/tpc.113.114769; https://doi.org/10.1038/msb.2011.97).

Reviewer 2 Report

Comments and Suggestions for Authors

Comments to the authors,

In the present work, the authors performed ribosome profiling (Ribo-seq) in maize to identify the ribosome pausing during etiolated maize seedlings exposed to light. This study uncovered the translational regulation of environmental stimulus. Overall, this is a meaningful work. But there were several points still need to be addressed before accepted. My decision is "Major revision".

1. The quality of Figure 3 need to be much improved.

2. The gene ID should be italicized throughout the text. “Arabidopsis” also need to be in italic font.

3. The English writing still need to be much improved.

4. The discussion section should not be a duplication of the results section. It is no need to cite figures in discussion section.

Comments on the Quality of English Language

The English writing still need to be much improved.

Author Response

Comment:In the present work, the authors performed ribosome profiling (Ribo-seq) in maize to identify the ribosome pausing during etiolated maize seedlings exposed to light. This study uncovered the translational regulation of environmental stimulus. Overall, this is a meaningful work. But there were several points still need to be addressed before accepted. My decision is "Major revision".

1.The quality of Figure 3 need to be much improved.

Response:

Thank you very much for review our manuscript. We have considered these comments carefully and tried our best to address every one of them. We made modifications on Fig. 3. This figure contains lots of information. To facilitate the explanation, we labeled different types of genes on each panel. The previous version looks crowded and was hard to read. In the revised figure, we made the following changes to make it more concise and clearer. First, we changed the colors of some labeled gene names to better discriminate from the background. Second, we adjusted the positions of some gene names on each panel to separate them from each other.

2.The gene ID should be italicized throughout the text. “Arabidopsis” also need to be in italic font.

Response:

We changed all the gene ID into italic fonts according to the reviewer’s suggestion. We did not change the font of “Arabidopsis” because it is common nouns similar to rice, wheat, maize and so on. Their Latin name such as “Arabidopsis thaliana”, “Oryza sativa L.” and “Zea mays L.” need to be italic.

3.The English writing still need to be much improved.

Response:

We asked a commercial agency (Planteditors, https://planteditors.com/) to polish our language for the previous manuscript. The company is located in Chicago, and the editors are all native English speakers and most of them earned Ph.D degree in plant sciences. In the revised manuscript, we also asked them for the improvement of the grammars.

4.The discussion section should not be a duplication of the results section. It is no need to cite figures in discussion section.

Response:

We removed the figure citations in the discussion of the revised manuscript accordingly.

Reviewer 3 Report

Comments and Suggestions for Authors

The manuscript titled “ Ribosome pausing negatively regulates protein translation in maize seedlings during dark-to-light transitions”, explores the regulation of translation in etiolated maize seedling exposed to light using ribosome profiling (Ribo-seq). It identifies over 400 ribosome-pausing events that negatively regulate translation, which are released upon illumination. The discovery of a conserved nucleotide motif downstream of pausing sites provides insight into translation regulation mechanisms and potential targets for controlling plant gene expression. This is a novel aspect in a field of intense research. Still some issues need to be clarified, as listed below, before the manuscript can be accepted for publication in I.J.M.S..

1.      For the introduction part, could you elucidate the primary objective of this study and discuss its significance within the current landscape of research on translational regulation in plants?

2.      For the experiment part, why were the time points of 0.5, 1, and 2 hours chosen, given that they are not equally spaced intervals? Why not select time intervals that increase equally to better compare the differences in experimental results?

3.      The study reports the identification of a conserved nucleotide motif downstream of the pausing sites. Could you elaborate on the potential significance of this motif and its implications for the regulation of translation?

4.      Considering your findings, what future research directions or next steps do you propose to further advance this area of study?

5.      Could you elaborate on the ribosome profiling technique utilized in this study, highlighting any specific encountered and the strategies implemented to overcome them?

Author Response

Comment:

The manuscript titled “Ribosome pausing negatively regulates protein translation in maize seedlings during dark-to-light transitions”, explores the regulation of translation in etiolated maize seedling exposed to light using ribosome profiling (Ribo-seq). It identifies over 400 ribosome-pausing events that negatively regulate translation, which are released upon illumination. The discovery of a conserved nucleotide motif downstream of pausing sites provides insight into translation regulation mechanisms and potential targets for controlling plant gene expression. This is a novel aspect in a field of intense research. Still some issues need to be clarified, as listed below, before the manuscript can be accepted for publication in I.J.M.S..

1.For the introduction part, could you elucidate the primary objective of this study and discuss its significance within the current landscape of research on translational regulation in plants?

Response:

Thank for the suggestion. We made the revisions in the introduction section accordingly as the followings:

1)We emphasized the primary objective of this study in line 94-98 in the revised manuscript:

“Although recent progress indicates that translational regulation is a general and important mechanism by which plants respond to various developmental and environmental signals, whether plants undergo ribosome pausing are currently poorly understood. This study aims to reveal the genes under the regulation of ribosome pausing and identify the sequence features triggering the ribosome pausing in plant.”

2) We discussed the significance of this study in line 110-113 in the revised manuscript:

“Our finding provided globally evidence that plant gene expressions were broadly regulated by ribosome pausing, and the identified nucleotide motif triggering the ribosome pausing could be utilized in artificial modulation of translation for plant transcript and the molecular breeding of crops.”

2. For the experiment part, why were the time points of 0.5, 1, and 2 hours chosen, given that they are not equally spaced intervals? Why not select time intervals that increase equally to better compare the differences in experimental results?

Response:

According to the results from bacteria and Arabidopsis, translational regulation is rapid response to environment changes. Light exposure is a strong environmental stimulus for etiolated seedlings, and we speculated that the translational regulation might occur during the early period, so we selected short and equally spaced time points at the early stage (0, 0.5 and 1h) to investigate the translational regulation events. The results in Figure 2 showed that translational regulation mainly occurred in 0.5h after light treatment, which verified our hypothesis. We continued to collect tissues at 2h and 4h time points because we intended to compare the translational and translational regulation on genes in the prolonged period.

3. The study reports the identification of a conserved nucleotide motif downstream of the pausing sites. Could you elaborate on the potential significance of this motif and its implications for the regulation of translation?

Response:

The GC-rich motif is highly enriched at the downstream of the pausing sites in RNAs, we hypothesized that the GC-rich motifs tended to form secondary structures and hinder the movement of ribosomes along mRNAs. Therefore, the motif may function as a knob to control the translation rates of genes. The importance of the secondary structures in translational regulation were also revealed by a recent study in Arabidopsis, which demonstrated the presence of secondary structure downstream of uORF determined the selection of ATGs between uORF and mORF by ribosomes, and status of the secondary structures were affected by the activities of RNA helixes (https://doi.org/10.1038/s41586-023-06500-y). We speculated the formation or loosening of the secondary structure of the GC-rich motif in our finding might also affected by the certain RNA helix genes, and the translation rate of the corresponding RNAs could be modulated with the status changes of the secondary structure of the GC-rich motif in certain conditions.

The finding of the conserved motif has potentials for the artificial control of gene expression through translational regulation. In bacteria, Pro-Pro rich peptide motifs were found to be related to ribosome pausing (https://doi.org/10.1016/j.celrep.2015.03.014). The peptide motif is hard to be used in artificial control of gene translation, because amino acid changes easily change the gene function. However, alterations in nucleotide motif do not necessarily cause amino acid changes, therefore, the motif in our findings could be readily used as the target to artificially control the translation rate of interesting genes in biological studies or crop molecular breeding, although more investigations need to be conducted to test this possibility.

4. Considering your findings, what future research directions or next steps do you propose to further advance this area of study?

Response:

Although our discovery is interesting and exciting, it is only a phenomenon and there is still a long way to go before fulling understanding the mechanisms behind the ribosome pausing and the potential application of the GC-rich motifs in artificial control of translation of genes. We have designed some experiments for further research. First, we could fuse a reporter protein such as GFP with the conserved or synonymously mutated GC-rich motifs, and transform them into maize plants. By comparing the GFP protein levels between two different types of transgenic plants, we could verify the real function of this motif in protein translation in vivo. Second, we will examine whether the ribosome pausing and the conserved motif is a widely spread phenomena in plants facing various environmental changes. We will test this for plants under bio- and abio-stresses. Third, we will select several disease resistance genes containing the conserved GC-rich motif, and conduct synonymous mutation in the GC-rich motif by gene editing to improve the disease resistance of crops.

5. Could you elaborate on the ribosome profiling technique utilized in this study, highlighting any specific encountered and the strategies implemented to overcome them?

Response:

We performed ribosome profiling in this research following the described methods (https://www.science.org/doi/10.1126/science.1168978 and https://doi.org/10.1073/pnas.1614788113). These steps include isolation of ribosome protected fragments, end repairing, adapter addition, circularization, rRNA deletion, and library amplification. A critical procedure in Ribo-seq is the removement of huge amount of rRNAs from the RNase-digested ribosome samples. With the plant ribozero rRNA removing kit used in the literature, we could only deplete less than 10% of rRNAs. To overcome this problem, we designed rDNA probes complementary to the most abundant rRNA reads in our Ribo-seq data. With these home-designed probes we were able to remove 70%-90% of rRNAs from the input RNAs, thus efficiently improved the valid reads.

Another improvement in this study is the introduction of the parameter of translation intensity (TI) to authentically reflect the translation progression of RNAs. We first used the parameter translational efficiency (TE), which has been widely used in publication, but failed to reveal good relationships between TE and pausing score. Then we take the ribosome coverage into considerations by introducing the parameter of TI, and successively revealed the high coordinates between TI and ribosome pausing. 

Round 2

Reviewer 2 Report

Comments and Suggestions for Authors

NA

Author Response

Comments and Suggestions for Authors

NA

Response:

thank you for processing our manuscript again.